# Prognostic Factors and Biomarkers of Responses to Immune Checkpoint Inhibitors in Lung Cancer

**DOI:** 10.3390/ijms20194931

**Published:** 2019-10-05

**Authors:** Andrea Bianco, Fabio Perrotta, Giusi Barra, Umberto Malapelle, Danilo Rocco, Raffaele De Palma

**Affiliations:** 1Department of Translational Medical Sciences, University of Campania “L Vanvitelli”, 80131 Naples, Italy; 2Department of Pneumology and Oncology, A.O. dei Colli, Hosp. V Monaldi, 80131 Naples, Italy; danilorocc@yahoo.it; 3Department of Medicine and Health Sciences “Vincenzo Tiberio”, University of Molise, 86100 Campobasso, Italy; fabio.perrotta@unimol.it; 4Department of Precision Medicine, University of Campania “L Vanvitelli”, 80131 Naples, Italy; giusi_barra@hotmail.it (G.B.); raffaele.depalma@unicampania.it (R.D.P.); 5Department of Public Health, University of Naples “Federico II”, 80131 Naples, Italy; umbertomalapelle@gmail.com

**Keywords:** lung cancer, immune checkpoint inhibitors, tumor microenvironment, immune evasion, Treg, nivolumab, pembrolizumab, atezolizumab, durvalumab

## Abstract

Manipulation of the immune response is a game changer in lung cancer treatment, revolutionizing management. PD1 and CTLA4 are dynamically expressed on different T cell subsets that can either disrupt or sustain tumor growth. Monoclonal antibodies (MoAbs) against PD1/PDL1 and CTLA4 have shown that inhibitory signals can be impaired, blocking T cell activation and function. MoAbs, used as both single-agents or in combination with standard therapy for the treatment of advanced non-small cell lung cancer (NSCLC), have exhibited advantages in terms of overall survival and response rate; nivolumab, pembrolizumab, atezolizumab and more recently, durvalumab, have already been approved for lung cancer treatment and more compounds are in the pipeline. A better understanding of signaling elicited by these antibodies on T cell subsets, as well as identification of biological determinants of sensitivity, resistance and correlates of efficacy, will help to define the mechanisms of antitumor responses. In addition, the relevance of T regulatory cells (Treg) involved in immune responses in cancer is attracting increasing interest. A major challenge for future research is to understand why a durable response to immune checkpoint inhibitors (ICIs) occurs only in subsets of patients and the mechanisms of resistance after an initial response. This review will explore current understanding and future direction of research on ICI treatment in lung cancer and the impact of tumor immune microenvironment n influencing clinical responses.

## 1. Introduction

Immunotherapy has marked a revolution in the treatment of lung cancer. By manipulating the immune response to target tumour cells, disease control is achievable in subsets of lung cancer patients, resulting in prolonged survival, although resistance after the initial response is not uncommon. 

Scientific interest is now directed to understanding the mechanisms of response to immune checkpoint inhibitors (ICIs) and determining prognostic factors and biomarkers.

Immune checkpoints, physiologically, are involved in preventing T cells from turning against healthy tissue through inhibitory and stimulatory pathways. To evade the immune response, cancer cells may develop the ability to exploit those pathways, impairing specific T cell activation that may lead to the mounting of a specific anticancer response [1]. Major targets for approved pharmacological interventions are the surface molecules PD-1 (programmed cell death protein-1) and CTLA-4 (cytotoxic T-lymphocyte associated antigen-4). Additional targets, including LAG-3, TIM-3, TIGIT, VISTA, B7/H3, OX40, ICOS, GITR, 4-1BB and CD40, are potential candidates for the development of new therapeutic options.

The cell-surface receptor PD-1 is expressed by T cells on activation during priming or expansion and binds to one of two ligands, PD-L1 or PD-L2. Many cell types can express PD-L1, including tumor cells and immune cells after exposure to cytokines, such as interferon (IFN)-γ; however, PD-L2 is expressed mainly on dendritic cells in normal tissues [2]. It must be considered that PD1 and CTLA4 can be expressed in a dynamic fashion on different T cell subsets; therefore, an approach aiming to tailor a precise therapy should be undertaken, characterizing the phenotypic expressions of those molecules on T cells in different patients. Binding of PD-L1 or PD-L2 to PD-1 generates an inhibitory signal that attenuates the activity of T cells. When inactivation of T cells occurs through such mechanisms, cancer cells evade attack. Immune checkpoint inhibitors (ICIs) have been developed to target those self-tolerance pathways that are exploited by tumors to escape immune recognition and destruction (Figure 1). Monoclonal antibodies (MoAbs) against PD1/PDL1 and CTLA4 have shown that inhibitory signals can be impaired blocking T cell activation and function. Among ICIs, nivolumab and pembrolizumab target the checkpoint protein PD-1, whereas atezolizumab and durvalumab target PDL1, all approved by FDA for treatment of lung cancer patients. Ipilimumab targets CTLA-4 and is awaiting approval for lung cancer treatment [3,4,5]. Beyond the expression of PD-L1, other biomarkers are under investigation to better stratify patients who may benefit from immune-checkpoint inhibitors (ICIs) (Table 1). One of these is represented by human leukocyte antigen class I (HLA-I). Homozygosity in at least one HLA-I locus (“A,” “B” or “C”) has been associated with a reduction of overall survival after treatment with ICIs [6]. Another promising biomarker is LAG-3, which has showed a synergy with PD-1 in different cancer types [7]. Conversely, STK11/LKB1 inactivation represents a negative predictive biomarker of response to ICIs [8]. In order to consolidate antitumor responses, signaling elicited by these antibodies on T cell subsets need to be better understood; the possible role of these molecules in regulating Treg function in cancer is an area of growing interest. A large body of research is now focused on tumor microenvironment (TME) and its influence on cancer growth and immune evasion mechanisms, which in turn affect responses to immune checkpoint inhibitors. In that context as well, tumor infiltrating lymphocytes (TIL) (CD4+ and CD8+) are emerging as potential independent prognostic factors for overall survival (OS) and response rate (RR) regarding ICI treatment [9,10]. Molecules targeting factors of tumors’ microenvironments, including IDO1 or TLR, are under investigation. Furthermore, research has shown that commensal microbiota can also affect ICIs’ responses in a positive or negative manner [11]. This article provides an overview of clinical impact of immune checkpoint inhibitors on lung cancer management, in light of current knowledge on immune pathways in terms of biological determinants of sensitivity/resistance and correlates of efficacy.

## 2. The Clinical Implications of ICIs in Non-Small Cell Lung Cancer (NSCLC) 

Since the FDA approval of nivolumab in 2015, immune checkpoint inhibitors have deeply shaped metastatic and locally advanced NSCLC treatment algorithms. According to ASCO guidelines, to date, three different drugs are presently FDA approved and recommended in clinical practice for the treatment of metastatic NSCLC: nivolumab, pembrolizumab and atezolizumab; specifically, pembrolizumab is recommended for monotherapy, and in the first line setting for patients with nonsquamous or squamous NSCLC without driver gene mutations (EGFR/ALK) and with a high PD-L1 expression (tumor proportion score (TPS) ≥ 50%). Nivolumab or atezolizumab are recommended in a second-line setting for nonsquamous or squamous NSCLCs regardless of PD-L1 expression. Lastly, pembrolizumab, can be employed as a second-line treatment in nonsquamous or squamous NSCLC patients with a PD-L1 expression ≥1% [12,13,14,15,16,17,18,19,20]. 

Regarding locally advanced NSCLC, durvalumab currently represents the only FDA approved and recommended ICI for the treatment of unresectable stage III NSCLC patients, irrespective of histological type and PD-L1 expression, whose disease has not progressed after a previous chemoradiotherapy treatment [21]. All four of these ICISs exert their activity by blocking either PD-1 (nivolumab and pembrolizumab) or PD-L1 (atezolizumab and durvalumab), interfering with and partially preventing tumor cells’ immune-escape, thus enhancing patients’ immune surveillance and T-cell-mediated responses to cancer cells [22,23,24]. A promising line of research in the treatment of metastatic NSCLC is exploring chemo-immunotherapy combination therapies, which may represent a potential standard-of-care therapy, replacing chemotherapy alone. A series of groundbreaking trials—although not yet ASCO-recommended—have led to FDA approval for pembrolizumab and atezolizumab in naive NSCLC patients, irrespective of PD-L1 expression: combined with pemetrexed + platinum (nonsquamous histology)/(nab)paclitaxel + carboplatin (squamous histology) and with (nab)paclitaxel + carboplatin + bevacizumab (nonsquamous histology), respectively [20,25,26,27,28] (Table 2).

## 3. ICIs and Special Populations: Oncogene-Addicted Patients

One key exclusion criterion in first-line treatments with immune checkpoint inhibitors, both as single agent treatments and as combination therapies, has been represented by the expression of driver gene mutations (EGFR/ALK). Indeed, most subgroup analyses intending to show a survival benefit favoring ICI treatment over standard chemotherapy in tirosyne kinase inhibitor (TKI)-pretreated patients, has failed to meet that endpoint [29,30,31]. The scientific rationale underlying these results may be represented by the lower PD-L1 expression and lower mutational burden expressed in oncogene-addicted patients [29]. Consequently, in oncogene addicted patients, the tumor growth may be sustained within an uninflamed tumor microenvironment with weak immunogenicity. Dong et al. [32] analyzed 255 resected NSCLC samples, documenting a significantly higher proportion of strongly positive PD-L1 expression within the EGFR wild-type group than in the mutated patients. Furthermore, a decreased mutation burden, lack of T-cell infiltration and reduction in proportion of PD-L1+/CD8+ TILs was observed in EGFR mutated samples. However, things could be about to change as a result of interesting data from some recent trials. In cohort 1 of the ATLANTIC (advanced non-small-cell lung cancer) study, a single arm phase II trial—111 EGFR+/ALK+ NSCLC-affected patients with ≥25% (74 patients) or <25% (37 patients) PD-L1 expression progressing after at least two previous lines of treatment (but ICIS-treatment naive) received durvalumab, an anti-PD-L1 antibody, for up to one year; as a result, in 12.2% of patients with ≥25% PD-L1 expression, an objective response was noted, as were favorable safety and tolerability profiles [33]. More importantly, with reference to the above-mentioned IMpower150 study, a protocol amendment was added to its original design in order to include EGFR+/ALK+ NSCLC-affected patients; remarkably, in that subgroup of patients as well, results managed to be consistent with those of the intention-to-treat population. In fact, once again, results favored the four-drugs arm, including atezolizumab, over the standard one (carboplatin + paclitaxel + bevacizumab), granting a median progression-free survival (mPFS) of 9.7 months compared to 6.1 months (hazard ratio (HR): 0.59) [28]. Based on those results, atezolizumab has been approved for patients who develop resistance to specific TKIs [34]. Phase I trials documented a reasonable safety profile when ICIs were combined with EGFR a mutation directed TKI [35,36,37]. Several clinical trials investigating the combination of a TKI with an ICI are currently recruiting (e.g., NCT02364609, NCT01454102 and NCT01998126).

## 4. Current Biomarkers in Immune Checkpoints Inhibitors 

### 4.1. PD-L1 Expression and TMB

In order to correctly administer immunotherapy, the National Comprehensive Cancer Network (NCCN) and the College of American Pathologists (CAP), the International Association for the Study of Lung Cancer (IASLC), and the Association for Molecular Pathology (AMP) guidelines strongly recommended that the expression of PD-L1 protein should be tested in patients [38,39]. As a result of the phase III KEYNOTE-024 clinical trial, the monoclonal human IgG4 antibody, pembrolizumab, directed against PD-1, has to be considered the first line treatment choice in advanced NSCLC patients with an expression of PD-L1 in tumor cells (tumor proportion score, TPS) ≥50% [40]. However, the actual relevance of the levels of PDL-1 expression in tumor samples as indicators of ICIs’ responses is under debate.

Recent evidence from the phase III KEYNOTE-042 trial strongly encourages the adoption of pembroluzimab as an alternative to platinum-based therapy, even in cases with low PD-L1 (≥1%) expression. To date, in addition to pembrolizumab, other drugs, such as nivolumab (anti-PD-1), and anti-PD-L1 atezolizumab and durvalumab (anti-PD-L1), are clinically available; for each different drug there is a related anti PD-L1 antibody clone for the immunohistochemistry (IHC) or immunocytochemistry (ICC) evaluation of TPS. In particular, the 22C3 clone (Dako, Carpinteria, CA, USA) was used for pembrolizumab treatment choice, whereas the clones 28-8 (Dako), SP142 (Ventana, Tucson, AZ, USA) and SP263 (Ventana) were related to the administration of nivolumab, atezolizumab and durvalumab, respectively [41]. The 22C3 clone (Dako) associated with pembrolizumab has been the only test approved by Food and Drug Administration as a “companion diagnostic” [42]. Due to the large number of assays and immunostaining platforms available, several efforts were made in order to harmonize and standardize the results obtained in the TPS evaluation. In particular, the Blueprint project showed a high concordance rate in TPSs for three (22C3, 28-8 and SP263) out of four clones [41]. These results were confirmed in the second phase of the Blueprint project performed on routine clinical samples [43]. Potential limitations in the adoption of PD-L1 as a universal biomarker for immunotherapy response were represented by: (i) a constitutive expression of PD-L1 in tumor cells without evidence of a pre-stimulation by the immune cells; (ii) the heterogeneity of the different assays’ performances, the subjective interpretation performed by pathologists of TPSs and relative thresholds adopted, which may all lead to false positive or negative results; and (iii) the assessment of tumor infiltrating lymphocytes (TILs). [44] Taking into account the data coming from nivolumab (CheckMate 017 and Checkmate 057), pembrolizumab (KEYNOTE-010 and KEYNOTE-024), atezolizumab (OAK) and durvalumab (PACIFIC) pivotal trials, it clearly appears that PD-L1 must be considered as only a prognostic and not a predictive marker.

Another potentially important marker of response to ICIs has been identified in the tumor mutational burden (TMB). High TMB represents the presence within the tumor of a high number of non-synonymous genomic alterations, an expression of a genomic instability, with the potential to generate several neoantigens [45]. As shown by Rizvi et al., NSCLC patients with high TMBs, treated with pembrolizumab, showed higher objective responses (ORs) and progression-free survival (PFS), than those with low TMBs [46]. In addition, in the CHECKMATE-227 clinical trial, the association of two different immunodrugs (nivolumab plus ipilimumab) showed a similar improvement in PFS in NSCLC patients with at least 10 mutations per megabase [47] when compared to standard-of care chemotherapy; these results were irrespective of PD-L1 expression levels: ORR—45.3% versus 26.9%; mPFS—7.2 months versus 5.5 months; HR for disease progression or death—0.58 (*p* < 0.001). However, following a request by the EMA-CHMP (European Medicines Agency’s Committee for Medicinal Products for Human Use) and by the FDA about the OS analyses of that study, with reference to both the high-TMB and the low-TMB (<10 mutations per megabase) subgroups of patients, not only were the HRs for OS with nivolumab + ipilimumab versus chemotherapy comparable between the two subgroups (0.77 and 0.78, respectively), but the mOS data also favored nivolumab + ipilimumab over standard-of-care chemotherapy in both these subsets of patients (23.03 months versus 16.72 months and 16.20 months versus 12.42, respectively). More extensive studies are required to define the role of TMB as a prognostic biomarker [48]. An important limitation for testing TMB in clinical practice is represented by the requirement of a large tissue specimen availability for analysis. Recently, several studies have evaluated the possibility of adopting the so called “liquid biopsy” for the evaluation of blood-based TMB (bTMB), with the adoption of a highly sensitive approach in next generation sequencing (NGS). This approach showed a potential clinical benefit in NSCLC patients treated with anti–PD-1 and anti–PD-L1 drugs, but further investigations are needed. [49,50,51]

### 4.2. Neoantigens

Neoantigens are antigens generated from wild-type antigens due to somatic mutations, that can be recognized by the patients’ T cells via class I major histocompatibility complex (MHC I), but with a higher binding affinity when compared to the wild-type antigen/MHC I binding, apparently due to their enhanced immunogenicity [45]. In a recent trial, extrapolating data from the Cancer Genome Atlas, Ghorani and colleagues investigated whether the different binding affinities between wild-type and mutated antigens—also known as the differential agretopicity index or DAI—may represent a statistically significant prognostic response biomarker in stage III/IV NSCLC or melanoma affected patients. As a result, data regarding NSCLC patients showed thata low mean DAI was linked to worse OS (*p* = 0.004), in contrast to neoantigen mean DAI and neoantigen load, that were associated with improved OS (*p* = 0.04 and *p* = 0.023, respectively); interesting results for sure, but more studies and more patients will be needed to better interpret these data [52]. 

### 4.3. STK11 Mutations 

Another interesting prognostic biomarker may be represented by STK11 (serine/threonine kinase 11), one of the most mutated tumor-suppressor genes in NSCLC, that seems to be frequently associated with KRAS mutations. For example, in a recent retrospective analysis, including 302 stage III/IV NSCLC-affected patients, 25 of which were STK11-mutated, 13 out of the 25 presented a KRAS co-mutation (52%, *p* = 0.0008). Furthermore, although no significant correlation to a worse OS or PFS was found in STK11-mutated patients, a trend towards worse OS was noted in STK11/KRAS co-mutated patients [53]. A recent investigation not only confirmed the STK11/KRAS association, but also demonstrated that an STK11/KRAS co-mutation was associated with lower RR to ICI treatment and shorter OS and PFS (*p* < 0.001, *p* < 0.001 and *p* = 0.0015, respectively), suggesting the importance of assessing its potentially negative prognostic role in this subset of patients in further prospective trials [8]. Similar conclusions have been reported in another retrospective study evaluating 567 NSCLC-affected patients [54]. More recently, in non-squamous NSCLC treated with a combination of platinum, pemetrexed and pembrolizumab, the STK11/LKB1 genomic alterations were associated with shorter PFS (mPFS 4.8 m versus 7.2 m, HR 1.5, 95% CI 1.1 to 2.0; *p* = 0.0063) and shorter OS (mOS 10.6 m versus 16.7 m, HR 1.58, 95% CI 1.09 to 2.27; *p* = 0.0083) compared with STK11/LKB1-wild type tumors [55]. Similar results emerged from a genomic study which documented that, in advanced NSCLC, the absence of mutation in STK11, TP53 and KEAP1 was associated with longer OS [56].

## 5. TME-Associated Biomarkers

### 5.1. TILs

High levels of tumor infiltrating lymphocytes (CD4+ and CD8+) should be considered, in NSCLC, as an independent positive prognostic factor for OS and for higher RR to ICI treatment [57]. Accordingly, one of the most recent trials on this topic involving a cohort of 26 NSCLC patients reported that patients whose tumors had a CD8+ lymphocyte count under 886/mm^2^ showed low RR to ICIs treatment (16.7%, *p* = 0.046), while patients whose tumors had a CD8+ lymphocyte count between 886 and 1899/mm^2^ exhibited a high RR (60%, *p* = 0.017), and NSCLC patients harboring CD8+/CD4+ ratios lower than 2 had a low RR (13.3%), when compared to those harboring ratios higher than 2 (RR ranging between 43% and 50% (*p* = 0.035)). To date, however, further validation for this marker is still required [58]. 

### 5.2. IDO1

IDO1 (indoleamine 2,3-dioxygenases) is a tryptophan-catabolic enzyme, responsible for catalyzing the conversion of tryptophan into kynurenine, contributing to immune tolerability, due to the immunosuppressive activity of tryptophan metabolites: T-effectors and NK cells’ functions are blockaded, and Treg, DC and myeloid-derived suppressor cells (MDSC) activities are enhanced [45,59]; however, IDO1 can also be overexpressed in different cancers, NSCLC included, contributing to the immunosuppressive TME [60,61]. In an interesting trial, Botticelli et coll. evaluated the association between IDO1 activity and resistance to ICI treatment, taking into account data from 26 NSCLC patients and assessing the kynurenine/tryptophan ratio as a possible prognostic biomarker and a surrogate for IDO1 activity. They reported a statistically significant correlation between early PD (<3 months), and higher kynurenine/tryptophan ratios and quinolinic acid concentrations (respectively, *p* = 0.017 and *p* = 0.005), while conversely noting that patients with lower kynurenine/tryptophan ratios and quinolinic acid concentrations exhibited better PFS and OS (respectively *p* = 0.018 and *p* = 0.0005). Future, larger studies may shed further light on the role of TME and IDO1 and feasibility of the kynurenine/tryptophan ratio and feasibility of the kynurenine/tryptophan ratio as a prognostic marker with respect to ICI treatment [62]. 

### 5.3. The Impact of the Microbiome on Responses to ICIs

Metagenomic data has hypothesized that the gut microbiome modulates the tumor responses to both chemotherapy and immunotherapy, potentially representing a biomarker for patients treated with ICIs. The gut microbiome retains the potential to interfere with innate and adaptive immune responses through different mechanisms. Firstly, in patients treated with CTLA-4 and PD1/PD-L1 blockade, the CD11b+ dendritic cells’ mobilization of the lamina propria, induced by the gut microbiome, enhances the Th-1 response against *Bacteroides fragilis’* capsular polysaccharides [63]. Furthermore, the loss of microbial diversity limits the antigen presentation, enhancing effector T cells’ functioning in the periphery and the tumor microenvironment (TME) [64]. Different microbiome clusters have been identified in responder and non-responder melanoma patients treated with anti-PD1 or CTLA4 MoAbs. Interestingly, the mice inoculated by fecal transplantation with microbes of the responder patients resulted in enhanced levels of SIY-specific CD8+ T cells but not of FoxP3+CD4+ regulatory T cells. [65] The heterogeneity of microbial species’ distribution in relation to country of origin, host diet and lifestyle means those promising data are not uniform and should be interpreted in relation to the broad variability that exists [66]. 

## 6. Expanding the Landscape: Tumor Microenvironment and T-reg Modulation in NSCLC Cancerogenesis

The TME is crucial for the growth of cancer cells. In the lung, the TME consists of stromal cells, fibroblasts, extracellular matrix, endothelial cells and immune cells—like myeloid-derived suppressor cells (MDSCs), tumor associated macrophages (TAM), dendritic cells (DC), mast Cells (MC) and different subtypes of lymphocytes [67]. These cells may often act like “the fuel that feeds the flame” for a tumor’s growth [67,68]. As a consequence, the immune system might not be able to mount an efficient response against the antigenic landscape of a tumor itself, thus, be unable to prime an efficient immune response, per se. Well recognized mechanisms adopted for immune-escape include: (1) a lack of proper presentation of tumor associated antigens (TAA); (2) immune-suppression achieved through the secretion of immunosuppressive cytokines and chemokines by TME cells inducing the polarization of T cell lymphocytes lacking antitumor efficacy, or impairing T cell recruitment in the tumor; (3) recruiting cells of immune system that downregulate immune responses; (4) mechanisms of immune editing due to tumor mutations giving rise to epitopes that are not recognized by immune system [2]. The latter mechanism is a paradoxical effect of the immune response that “selects” tumor clones bearing mutations not recognized by immune system, thus evading the response [69]. Indeed, T cells may differentiate into distinct T helper subsets. Cytotoxic T cells and Th1 cells are well known to be able to eliminate cancer cells, although the anti-tumor effect of these subsets may be severely impaired by a TME and the expression of inhibitory molecules on cell surfaces. Other T cell subsets, including Th2, Th17 and T regulatory cells, may have ambivalent roles in cancer [2,70]. In that scenario, T regulatory cells (Treg) play a crucial role [71].

Due to the peculiar role of Tregs in the down regulation of immune responses, we should address the possible role of these molecules as targets to regulate Treg function in cancer. In recent years, much effort has been devoted towards counteracting the expression of inhibitory molecules, such as anti-CTLA-4 and PD-1, on T cell surfaces. However, despite such efforts, the rate of responses to these therapies in cancer patients is still far from being satisfactory. CTLA4 and CD28 compete for the same ligand and deliver opposite effects. Indeed, on the effector T cells, the binding of CTLA4 mediates inhibitory functions while binding to CD28 triggers T cell activation [72,73].

Treg cells, differing from effector T cells, constitutively express CTLA-4, and that contributes to their suppressive function. Although the depletion of Tregs through the binding of CTLA4 has been thought to be a main mechanism of action of such an antibody in immunotherapy, several lines of evidence show that there is no change of Treg number in tumors upon immunotherapy with antiCTLA4, despite the efficacy of the treatment [74]. Therefore, it is likely that the major effect of the antiCTLA4 therapy is the impairment of Treg activation, following CTLA4 engagement. As a consequence, the impairment of T cell functions leads to the reactivation of antitumor T cell responses previously downregulated by Treg activity. Treg cells preferentially bind to activating costimulatory molecules, inducing their downregulation on APC surface, thus contributing to reducing the activation of effector T cells [75,76,77]. Once activated, CTLA-4 can inhibit cell stimulation through different mechanisms; besides the competition with CD28 for binding to B7.1/B7.2, signaling through CTLA4 can also inhibit IL-2 mRNA production or promote the expression of molecules associated to cell cycle arrest or inhibit the nuclear factors factor κB (NF-κB), nuclear factor of activated T cells (NF-AT) and activator protein 1 (AP-1) [72,75]. In Treg cells, CTLA-4 seems to be associated with the upregulation of FoxP3; indeed, the activation of this marker increases TGF-β production, which in turn promotes FoxP3 expression [73]. This mechanism is also implicated in the conversion of peripheral CD4+CD25− naive T cells to CD4+CD25+ regulatory T cells, thus enhancing immune suppression [73,78]. PD-1 is known to inhibit T cells’ proliferation and effector functions and is also expressed on the Treg’s surface. Differing from effector cells, PD-1 expressed on Tregs contributes towards promoting immune suppression, stimulating their proliferation. Moreover, the PD-1 ligand, PD-L1, can induce Foxp3 expression [79]. PD-L1 is associated with the inhibition of AKT/mTOR pathways and expanding iTregs [79,80]. Acting differently from CTLA-4, PD-1 is activated during the effector phase of T cells’ activation; after TCR stimulation and the binding of PD-1 with PD-L1, the cytoplasmic tail of PD-1 is phosphorylated and recruits the phosphatases SHP-1 and SHP-2, which dephosphorylate the molecules downstream of the TCR, such as ZAP-70 and LCK. Within the TME, those mechanisms are associated with cancer proliferation due to their effects on signaling pathways determining cytokine production, and T effector cells’ proliferation and survival [81,82,83]. Besides PD-1 and CTLA-4, several other inhibitory molecules expressed on Tregs are candidates to be targeted to improve anticancer therapies. In the last few years, particular interest has been directed to the IC TIM-3 as a good target for new lung cancer immunotherapies [84,85,86]. This molecule plays a key role in the inhibition of Th1 responses and the expression of cytokines such as TNF and INF-γ. TIM-3+ Treg cells are associated with more aggressive forms of tumors, producing more IL-10, granzyme and perforin [85,86]. In the absence of TIM-3 ligand galectin-9, the five tyrosine residues of TIM-3 cytoplasmic tail are phosphorylated and bind HLA-B associated transcript 3 (Bat-3). In that condition, TIM-3 can also promote effector T cell activity. When galectin-9 binds to TIM-3 instead, bat-3 is released and promotes inhibitory signals associated with programmed cell death, the suppression of Th1 and Th17 responses and the induction of tolerance [87]. TIGIT is an ICI expressed on some Treg subsets and intervenes especially in the suppression of Th1 and Th17 lymphocytes. it was found that TIGIT-positive Tregs have a potent immunosuppressive action and seem to produce higher levels of IL-10 [39,41]. The exact pathway associated with the inhibitory activity of TIGIT is not completely clear. It is known that it has a cytoplasmic tail containing an ITIM and an ITT motif and their phosphorylation is correlated to T cell inhibition. When the ligand CD155 binds to TIGIT, the phosphatase SHIP1, associated with blocking the signal transduction of PI3K and MAPK pathways, is recruited [88]. TIGIT promotes immune suppression and interferes with T effector cells’ activation and proliferation, downregulating molecules that compose or regulate the TCR. It has been widely recognized that the expansion of Tregs in lung cancer correlates with a poor outcome [89], and, so far, how the available immunotherapy drugs may modulate Treg in cancer has not been approached. Figure 2 depicts functional effects of MoAb on molecules expressed on surface of Tregs cells.

## 7. Discussion

The identification of novel molecular targets and the manipulation of immunity is expanding the horizons of lung cancer management [5,81,82,83,84,85,86,87,88,89,90]. Despite evidence of durable responses, currently approved antibodies against the immune regulators CTLA4 and PD-1/PD-L1 in lung cancer are effective only in subsets of patients, and resistance after an initial response may occur. A better understanding of the biological determinants of sensitivity and resistance, and correlates of the efficacy and characterization of the antitumor immune response, is required. Evolving knowledge on tumors’ immune microenvironments has the potential to significantly impact clinical management of lung cancer. 

In this context, analyses of number of and functions of Tregs may be of critical importance; and a better understanding of how to modulate Treg function and expansion in cancer may be fundamental to improving immunotherapy in cancer. While several studies have assessed Tregs in different tumors, we still lack a comprehension of how we may modify Treg functions to improve immune responses against cancer. Tregs are indeed plastic and may gain effector functions, thus contributing to anticancer responses rather than promoting tumor growth [90]. Assessing the effects of immune checkpoint blockades on Tregs should further improve our ability to promote efficient immune responses in cancer immunotherapy. 

## Figures and Tables

**Figure 1 ijms-20-04931-f001:**
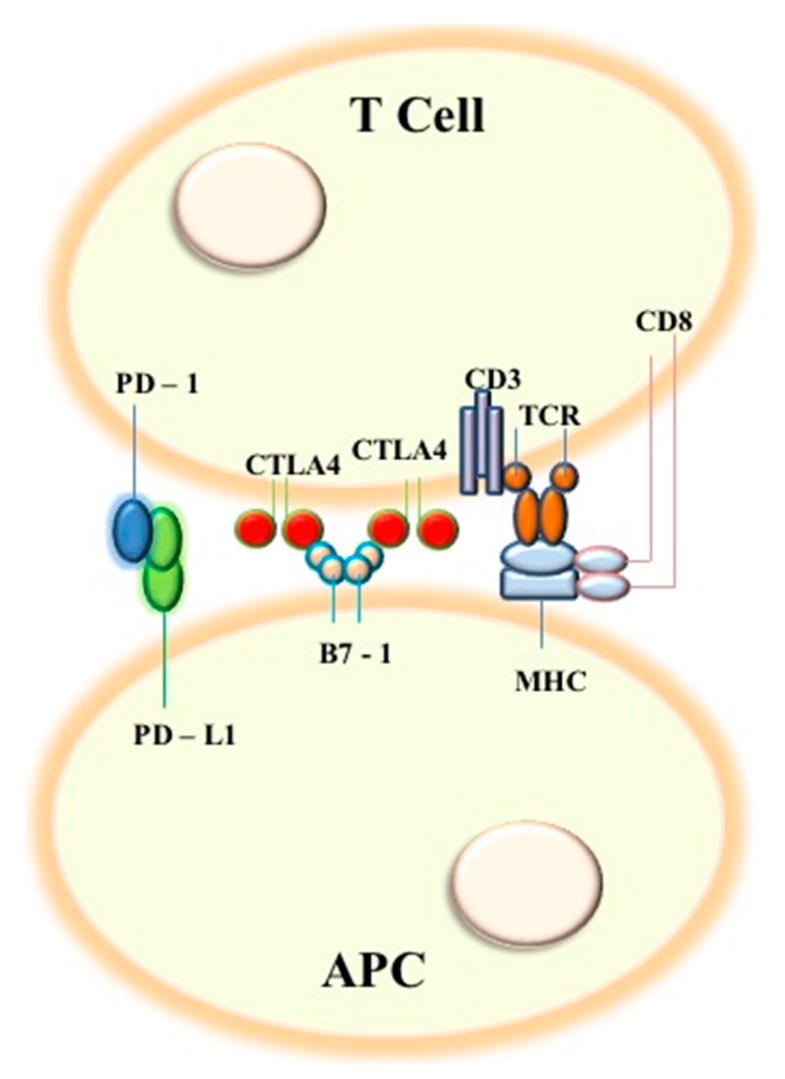
Schematic representation of principal interactions targeted by immune checkpoints inhibitors, including PD-1 (programmed cell death protein) and PD-L1 (programmed cell death ligand 1 protein), CTLA4 (cytotoxic T-lymphocyte antigen 4), B7-1, TCR and MHC (major histocompatibility complex) (with the co-function of CD3 and CD8 molecules) pathways, between T-lymphocytes (T cells) and antigen presenting cells (APCs).

**Figure 2 ijms-20-04931-f002:**
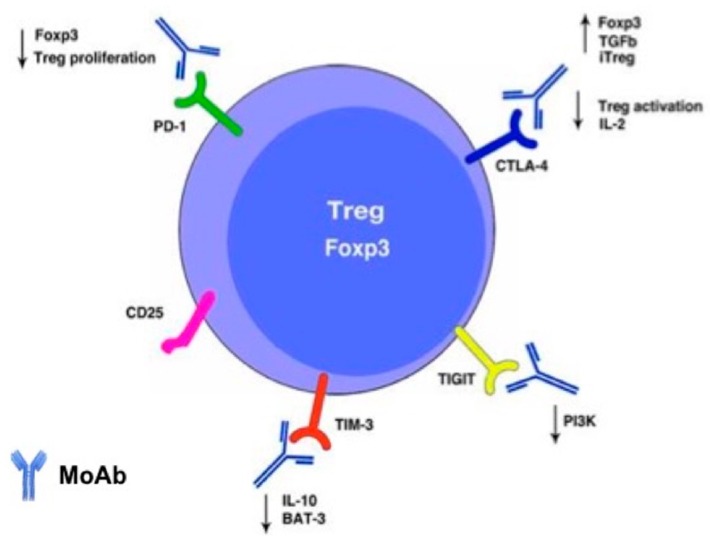
Functional effects upon the binding of monoclonal antibodies (MoAb) to molecules expressed on surface of Tregs cells. The binding of MoAbs with CTLA-4, PD1, TIGIT or TIM 3 induce changes in Treg functions, affecting the overall immune responses against tumors.

**Table 1 ijms-20-04931-t001:** Current and promising prognostic biomarkers in non-small cell lung cancer (NSCLC) treatment.

Prognostic Biomarker	Current State of Development
PD-L1 expression	FDA-approved and fully implemented in clinical practice
Tumor mutational burden	Under investigation
Differential agretopicity index	Under investigation
STK11 mutations	Under investigation
High levels of tumor infiltrating lymphocytes (CD4+, CD8+, CD8+/CD4+ ratio)	Under investigation
Kynurenine/tryptophan ratios	Under investigation
Quinolinic acid concentrations	Under investigation
Gut microbiome	Under investigation

**Table 2 ijms-20-04931-t002:** FDA-approved immune checkpoint inhibitors (ICIs) for the treatment of stage III/IV NSCLC.

ICI Treatment	Pivotal Trial	Setting	Target Population	FDA Approval
Nivolumab monotherapy versus docetaxel	CheckMate017	II line after chemotherapy failure	Stage III-B or IV Squamous NSCLC	March 2015
Nivolumab monotherapy versus docetaxel	CheckMate057	II line after chemotherapy failure	Stage III-B or IV Nonsquamous NSCLC	October 2015
Pembrolizumab monotherapy versus platinum- based chemotherapy	KEYNOTE-024	I line (PD-L1 ≥ 50%)	Stage IV Nonsquamous and squamous NSCLC	October 2016
Pembrolizumab monotherapy versus docetaxel	KEYNOTE-010	II line after chemotherapy failure (PD-L1 ≥ 1%)	Nonsquamous and squamous NSCLC	October 2016
Atezolizumab monotherapy	OAK	II line after chemotherapy failure	Stage III-B or IV Nonsquamous and squamous NSCLC	October 2016
Durvalumab monotherapy versus placebo	PACIFIC	Durvalumab after chemoradiotherapy	Stage III unresectable Nonsquamous and squamous NSCLC	February 2018
Pembrolizumab + cis/carboplatin + pemetrexed	KEYNOTE-189	I line	Nonsquamous NSCLC	August 2018
Pembrolizumab + paclitaxel/nab-paclitaxel + carboplatin	KEYNOTE-407	I line	Stage IV Squamous NSCLC	October 2018
Atezolizumab + carboplatin + paclitaxel + bevacizumab	IMpower 150	I line	Stage IV or recurrent metastatic Nonsquamous NSCLC	December 2018

ALK: anaplastic lymphoma kinase; CNS: central nervous system; ECOG PS: Eastern Cooperative Oncology Group Performance Status; EGFR: epidermal growth factor receptor; ICI: immune checkpoint inhibitor; NR: not reached; NSCLC: non-small cell lung cancer; OS: overall survival; PFS: progression free survival; TKI: tyrosine kinase inhibitor.

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
