# Peer review of "Prognostic Factors and Biomarkers of Responses to Immune Checkpoint Inhibitors in Lung Cancer"

_ijms, 2019, doi:10.3390/ijms20194931_

Round 1
Reviewer 1 Report
This review authored by Andra Bianco et al. is presenting the potential prognostic markers that could be used to predict responses to ICPI in lung cancer. Many of them are common to other cancers.
Comments:
1-They mention the ICPI that are approved for lung cancer with a large table of two pages giving information of OS, PFS, Setting… All this information depicted on the table is useful but does not go along with the aim of the publication and I think that it should be reduced. On the other hand it’s missing a table that could summarize the prognostic markers (PDL1 expression, tumor foreignness (neoantigens), etc .).
2-The legends within figure 1 are too small and blurry, they are unreadable.
3-Page 1, line 42: “… capability to exploit these pathways disrupting T-cell receptors’ (TCRs) ability to identify and eradicate tumour cells …” Not sure what they meant by this. It should be rewritten more clearly. The threshold of T-cell activation can be diminished by the co-inhibitor signals on T cells but the TCR specificity does not change.
4-Page 2, line 51: “ … We know that PD1 and CTLA4 are dynamically expressed on different T cell subsets that can either disrupt or sustain tumor growth. ” I can’t think of any way in which PD1 and CTLA4 signaling can disrupt tumor growth.
5- Page 2, lines 54-57: “Similarly, CTLA-4 …. T-cell response.” This paragraph is redundant and should be removed, as it is discussed later on.
6- Page 4, line 117. Please define TKI. Please explain the rationale of the exclusion criteria in patients’ expression gene mutation in EGFR/ALK.
7- Page 7, line 267. Antigen immunoediting should be mentioned as a mechanism of tumor immune scape. Chemokines not only affect the polarization of T lymphocytes but also the homing of these cells into the tumor.
8-Page 7, line 270: “Indeed, T helper cells may differentiate into...” Authors should remove the “helper cells” and just add that there are different types of T lymphocytes. Treg and Tc are not helper cells. Besides, there are also subtypes within the Tc and Treg.
9-Page 7, line 280: “CTLA-4 is analogous to CD28.” They are proteins with high homology but they are not analogous, they actually have opposite functions.
10-Page 7, line 285: “Due to this, the main effect of this antibody in immunotherapy is due to Treg impairment”. This claim is not clear; there is an important role of CTLA4 ICPI on priming de novo T cells that cannot be disregarded. There is no mention of the depletion of Treg induced by CTLA4 antibodies.
11- References are missing (e.g. line 73, 77, …., 293, 321).
Author Response
They mention the ICPI that are approved for lung cancer with a large table of two pages giving information of OS, PFS, Setting… All this information depicted on the table is useful but does not go along with the aim of the publication and I think that it should be reduced. On the other hand it’s missing a table that could summarize the prognostic markers (PDL1 expression, tumor foreignness (neoantigens), etc .).We have added a new table with the current prognostic biomarkers. About table 2 unfortunately OS and PFS were specific issues raised from a previous reviewer. We however feel that the core of this review is based on response biomarkers and consequently we reduced the table columns.
The legends within figure 1 are too small and blurry, they are unreadable.We wish to thank the reviewer. Now the legends within figure 1 are in bold.
Page 1, line 42: “… capability to exploit these pathways disrupting T-cell receptors’ (TCRs) ability to identify and eradicate tumour cells …” Not sure what they meant by this. It should be rewritten more clearly. The threshold of T-cell activation can be diminished by the co-inhibitor signals on T cells but the TCR specificity does not change.We agree with the Reviewer that this is a convolute sentence, we are aware that TCR specificity does not change after thymic education so that a T cell cannot variate antigen specificity. We regret to say that the misleading sentence has been due to the wrong writing by an English speaking native person who misled the profound sense of the sentence. We have corrected that in the text and apologize once again for this. “…ability to exploit these pathways impairing specific T cell activation that may lead to mount a specific anticancer response
Page 2, line 51: “ … We know that PD1 and CTLA4 are dynamically expressed on different T cell subsets that can either disrupt or sustain tumor growth. ” I can’t think of any way in which PD1 and CTLA4 signaling can disrupt tumor growth.Again , we agree with this Reviewer and apologize for the sentence that is written in an “obscure” way. We mean to state that PD1 and CTLA4 can be modulated on the surface of different T cell subsets, therefore expression (or lack of expression) of these molecules on T cell may modulate the T cell function promoting or inhibiting T cell activation. For instance, it is well known that the binding of CTLA4 on T cell surface with CD80 or CD86 present on the APC deliver an inhibitory signal to T cell. On the other hand, it is known from long time that CTLA-4 control activation of Treg (Wing K et al, Science, 322:271-275, 2008). In other words, activation of Treg through CTLA4 binding impairs activation of those cells that, upon activation, may impair T cell effector response. Therefore, since CTLA4 is constitutively expressed on Treg, the blocking of Treg may “release the break” of immune system, sustaining T cell activation in/versus the tumor. We apologize for the criptic sentence that we have rewritten. “It must be considered that PD1 and CTLA4 can be expressed in a dynamic fashion on different T cell subsets, therefore an approach aiming to tailor a precise therapy should be undertaken characterizing the phenotypic expression of these molecules on T cell in the different patient. For instance, the absence of PD1 on T cell surface in the tumor make unlikely a possible efficacy of anticancer treatment targeting this molecule.”
Page 2, lines 54-57: “Similarly, CTLA-4 …. T-cell response.” This paragraph is redundant and should be removed, as it is discussed later on.
We have removed this sentence according to the Reviewer suggestion.
Page 4, line 117. Please define TKI. Please explain the rationale of the exclusion criteria in patients’ expression gene mutation in EGFR/ALK.We have defined TKI and included the scientific basis of the reduced response to IPCIs in oncogene addicted patients based mainly on a uninflamed tumor microenvironment with a low immunogenicity.
Page 7, line 267. Antigen immunoediting should be mentioned as a mechanism of tumor immune scape. Chemokines not only affect the polarization of T lymphocytes but also the homing of these cells into the tumor.We agree with the Reviewer and we changed the sentence accordingly. Antigenic landscape of tumor is self and thus, per se, unable to prime an efficient immune response. Well recognized mechanisms adopted for immune-escape include: 1) lack of proper presentation of Tumor Associated Antigens (TAA); 2)immune-suppression achieved through the secretion of immunosuppressive cytokines and chemokines by TME cells exiting to polarization of T cell lymphocytes lacking antitumor efficacy, or impairing T cell recruitment in the tumor; 3) recruiting cells of immune system that downregulate immune responses; 4)mechanisms of immune editing due to tumor mutations giving rise to epitopes that are not recognized by immune system [2]. This latter mechanism is a paradox effect of immune response that “select” tumor clones bearing mutations not recognized by immune system thus evading the response. Indeed, T helper cells may differentiate into distinct T helper subsets. Cytotoxic T cells and Th1 cells are well known to be able to eliminate cancer cells although the anti-tumor effect of these subsets may be severely impaired by TME and expression of inhibitory molecules on cell surface. Other T cell subsets including Th2, Th17, and T regulatory cells may have ambivalent roles in cancer[2,66]. In this scenario, T regulatory cells (Treg) play a crucial role [67].
Page 7, line 270: “Indeed, T helper cells may differentiate into...” Authors should remove the “helper cells” and just add that there are different types of T lymphocytes. Treg and Tc are not helper cells. Besides, there are also subtypes within the Tc and Treg.
We have removed the ‘helper’ term which was misleading as suggested by the Reviewer
Page 7, line 280: “CTLA-4 is analogous to CD28.” They are proteins with high homology but they are not analogous, they actually have opposite functions.We apologize for this sentence. Obviously these are not analogous molecule. The right sentence is: CTLA4 and CD28 compete for the same ligand and deliver opposite effects. Indeed, on the effector T cells, the binding of CTLA4 mediate inhibitory functions while the binding to CD28 trigger T cell activation.
Page 7, line 285: “Due to this, the main effect of this antibody in immunotherapy is due to Treg impairment”. This claim is not clear; there is an important role of CTLA4 ICPI on priming de novo T cells that cannot be disregarded. There is no mention of the depletion of Treg induced by CTLA4 antibodies.We have rewritten the sentence according to the Reviewer suggestion: Although depletion of Treg through the binding of CTLA4 has been thought to be a main mechanism of action of this antibody in immunotherapy, several lines of evidences show that there is no change of Treg number in tumors upon immunotherapy with antiCTLA4, despite the efficacy of the treatment. Therefore, it is likely that the major effect of the anti CTLA4 therapy is the impairment of Treg activation following CTLA4 engagement. As consequence, the impairment of T cell functions lead to the reactivation of antitumor T cell reponse previously downregulated by Treg activity
References are missing (e.g. line 73, 77, …., 293, 321).The missing references have been added wherever needed.
Reviewer 2 Report
Overall the review is well organized.
1. Likely some more recent references have to be added. In particular:
Remon J, Ahn MJ, Girard N, Johnson M, Kim DW, Lopes G, Pillai RN, Solomon B, Villacampa G, Zhou Q. Advanced-Stage Non-Small Cell Lung Cancer: Advances in Thoracic Oncology 2018. J Thorac Oncol. 2019 Jul;14(7):1134-1155. doi: 10.1016/j.jtho.2019.03.022. Epub 2019 Apr 16. Review. PubMed PMID: 31002952.2. In particular, since January 2019 ICI therapy was approved also for patients with EGFR+/ALK+ mutations that previously failed TKI target therapy. This is a very important point to include and discuss in the review.
3. The most interesting point of the review is that described the effects of ICIs on Treg, in my opinion this part has to be stressed.
Minor points:
Please check al the abbreviations. moAbs appears the first time in the abstract without explanation change ICPIs with the most accepted ICIs, spell out it the first time in the text: line 39 and successively use the abbreviation the figures need a higher resolution check carefully typos
Author Response
Likely some more recent references have to be added. In particular:Remon J, Ahn MJ, Girard N, Johnson M, Kim DW, Lopes G, Pillai RN, Solomon B, Villacampa G, Zhou Q. Advanced-Stage Non-Small Cell Lung Cancer: Advances in Thoracic Oncology 2018. J Thorac Oncol. 2019 Jul;14(7):1134-1155. doi: 10.1016/j.jtho.2019.03.022. Epub 2019 Apr 16. Review. PubMed PMID: 31002952.
We have included this reference in the introduction
In particular, since January 2019 ICI therapy was approved also for patients with EGFR+/ALK+ mutations that previously failed TKI target therapy. This is a very important point to include and discuss in the review.We wish to thanks the reviewer for arising this point. We now discuss this point in the oncogene addicted paragraph.
The most interesting point of the review is that described the effects of ICIs on Treg, in my opinion this part has to be stressed.We agree the reviewer and we really appreciate that He/She highlighted that effects of ICIs on Treg may represent a novel finding in this field. In this version, this part has been extended (in some part rewritten) for making more readable the interaction between ICIs and Treg.
Minor points:
Please check all the abbreviations. moAbs appears the first time in the abstract without explanation change ICPIs with the most accepted ICIs, spell out it the first time in the text: line 39 and successively use the abbreviation the figures need a higher resolution check carefully typos
The abbreviations , typos and figures have been implemented.